# A Continuous Flow-through Microfluidic Device for Electrical Lysis of Cells

**DOI:** 10.3390/mi10040247

**Published:** 2019-04-13

**Authors:** Ying-Jie Lo, U Lei

**Affiliations:** Institute of Applied Mechanics, National Taiwan University, Taipei 10617, Taiwan; d93543008@ntu.edu.tw

**Keywords:** electrical cell lysis, continuous flow-through device, human red blood cells, Maxwell stress

## Abstract

In contrast to the delicate 3D electrodes in the literature, a simple flow-through device is proposed here for continuous and massive lysis of cells using electricity. The device is essentially a rectangular microchannel with a planar electrode array built on its bottom wall, actuated by alternating current (AC) voltages between neighboring electrodes, and can be incorporated easily into other biomedical systems. Human whole blood diluted 10 times with phosphate-buffered saline (about 6 × 10^8^ cells per mL) was pumped through the device, and the cells were completely lysed within 7 s after the application of a 20 V peak-to-peak voltage at 1 MHz, up to 400 μL/hr. Electric field and Maxwell stress were calculated for assessing electrical lysis. Only the lower half-channel was exposed to an electric field exceeding the irreversible threshold value of cell electroporation (*E*_th2_), suggesting that a cross flow, proposed here primarily as the electro-thermally induced flow, was responsible for bringing the cells in the upper half-channel downward to the lower half-channel. The Maxwell shear stress associated with *E*_th2_ was one order of magnitude less than the threshold mechanical stresses for lysis, implying that an applied moderate mechanical stress could aid electrical lysis.

## 1. Introduction

Cell lysis, breaking down the cell envelope (membrane, and wall if it exists) and releasing its contents, is an important step for many biomedical applications and has been studied in detail for decades [1,2,3]. Effective physical, chemical, and biological methods have been developed, from small (laboratory-scale; single-cell lysis) to large-scale (process-scale), for different applications. Selection of the most appropriate lysis method depends on the downstream cell lysate processing and analysis as well as different types of cells and applications. Here we consider electrical lysis [4].

When cells are subject to a sufficiently high electric field, exceeding a threshold value, *E*_th1_, the permeability of the cell membrane is increased such that materials (chemicals, drugs, genes, etc.) can be introduced into or taken out from the cells. The cell membrane recovers, blocking the material flow again, when the electric field is turned off [5]. When the applied electric field exceeds an even higher threshold value, *E*_th2_, the membrane will not recover and the cell is ruptured [6,7]. The above reversible and irreversible processes are referred to as electroporation in the literature, and they have significant biomedical applications. *E*_th1_ and *E*_th2_ are called the reversible and irreversible thresholds of electroporation, but they have not been clearly defined. *E*_th2_ is directly related to cell lysis via electricity; it varies with different electrical excitations, different cells, and different environments. Some related figures from experiments in the literature are cited below for illustration. 

Consider first the cells in suspension (without flowing) or in an adhesive state, subject to electric impulse excitation. *E*_th1_ and *E*_th2_ of mouse lyoma cells were found to be around 6 × 10^5^ V/m [5] and 10 × 10^5^ V/m [6], respectively, for an electric impulse with 5–10 μs duration; and three impulses of 8 × 10^5^ V/m with 5 μs durations were optimized for transferring genes into the cell. Survival rate of *Escherichia Coli* (*E. Coli*) could be reduced from 100% to 1% if the amplitude of the applied electric impulses (10 pulses of 20 μs, total exposure time 200 μs) is increased from about 7 × 10^5^ to 17 × 10^5^ V/m. However, the survival rate could also be reduced from about 80% to 10% if the total exposure time is increased from 10 μs to 400 μs when the field magnitude is fixed at 14 × 10^5^ V/m [7]. Adherent rat basophilic leukemia (RBL) cells were lysed in 30 ms for an electric impulse with an amplitude of 20 × 10^5^ V/m and duration of 1 ms, but a much longer time, about 1 s, was required if either the field amplitude was reduced to 5 × 10^5^ V/m or the time duration was reduced to 0.1 ms [8]. Human breast cancer cells, MCF-7 and HMT-3522, were lysed when subjected to a pulse at 20 × 10^5^ V/m with 10 μs duration [9]. Even aggregated cells can be killed. Twelve out of thirteen tumors (92%) were ablated using irreversible electroporation, by delivering across the tumor 80 pulses of 100 ms at 0.3 Hz with an electric field magnitude of 2.5 × 10^5^ V/m [10]. In summary, *E*_th2_ is not a definite value; cells could be lysed when they are subject to electric impulse(s) with a sufficiently high strength (say, on the order of 10^6^ V/m), provided they are exposed to the electric field for a sufficiently long time duration (say, on the order of ms). 

Consider next the situation when cells are subject to a continuous applied electric field. Acute myeloid leukemia (AML) cells were electroporated under a field of 3 × 10^4^ V/m in a contracted microchannel (capillary contracted from 20 μm to 10 μm), where the cells were also subject to mechanical stresses [11]. As the cell membrane remained largely intact, the value of 3 × 10^4^ V/m is more likely to be a value between *E*_th1_ and *E*_th2_. Jurkat cells were completely lysed within 33 ms under an alternating current (AC) field with 9 × 10^4^ V_pp_/m at 75 Hz, where V_pp_ refers to peak-to-peak voltage [12]. The small value of 9 × 10^4^ V_pp_/m (*E*_th2_) in comparison with those in the literature was attributed to the hypotonic effect associated with surfactant in the buffer [1]. Human red blood cells were completely lysed (ruptured) when they were driven by an electro-osmotic flow through an orifice in a microchannel, where the field strength was 1.2 × 10^5^ V/m [13]. In summary, the cells are in general subject to a much longer exposure time in a continuous applied electric field, and the field strengths required (of order 10^5^ V/m) were substantially less than those subject to impulsive excitations. 

It would be convenient and helpful for many biomedical applications if the cells in a flowing stream are lysed continuously as they flow through a lysing section, and the lysate moves subsequently downstream for further processing and analysis in a lab-on-a-chip device. The device in Ref. [13] mentioned above is a type of such a device, but the throughput is limited. In order to increase the throughput of the lysing device/component, much effort was spent by increasing the electric field strength while keeping the applied voltage sufficiently low. Lu et al. [14] proposed a system, consisting of four parallel microchannels with a height of 50 μm and 3D sawtooth electrodes built on opposite vertical side walls, for lysing human colorectal cancer HT-29 cells in buffer by applying AC voltages less than 10 V_pp_. There was a total of 180 sawteeth for each channel, with a minimum spacing of 30 μm, maximum spacing of about 116 μm, and a pitch of 90 μm. There were 10^5^–10^6^ cells per mL in buffer, and they were pumped through the device at 0.25–1 μL/min. Seventy-four percent of the cells were lysed under a field with a characteristic strength of 2.83 × 10^5^ V_pp_/m (for 8.5 V_pp_ across 30 μm) at 10 kHz as the cells and buffer flowed through the channels. Note that the electric field felt by a cell is not uniform; it varies approximately between 0.73 × 10^5^ V_pp_/m (8.5 V_pp_ across 116 μm) and 2.83 × 10^5^ V_pp_/m. Shihini and Yeow [15] proposed a rectangular channel (length 1 cm, width 1 mm, and height 75 μm) sandwiched by two plate electrodes from top and bottom, with the lower stainless steel electrode coated with multi-walled carbon nanotubes (MWCNTs) with lengths of 6–8 μm and diameters of 2–6 μm at a surface density of 10^4^ cm^−2^, for generating a local high intensity field next to the lower electrode. *E. Coli* cells were completely lysed by applying 40 V_pp_ (i.e., a field of 5.3 × 10^5^ V/m) and 85 V_pp_ (11.3 × 10^5^ V/m) AC voltages, respectively, for cases with and without an MWCNT coating, under a flow rate 167 μL/min. The throughput can be doubled to 333 μL/min using a 75 V_pp_ voltage with an MWCNT coating. Lower applied voltages for cell lysis could probably be expected if the random orientated MWCNT layer was replaced by the self-aligned highly ordered 3D structures of aluminum nanospike arrays (spike height = 350, 700, or 1100 nm, pitch = 1.2 μm), as human cervical (HeLa) cancer cells were lysed when subjected to AC pulses with an amplitude of 2 V_pp_ and a duration of 12 ms in a stagnant (not flowing) chamber with a height of 100 μm [16]. However, the high intensity local field emitted from a spike can penetrate distances on the order of microns, which is in general much less than the height of a typical microchannel, as shown in a calculation in [16]. The low value of 2 V_pp_ for cell lysis in [16] is probably associated with the situation where cells settle on the spikes in a “stagnant” chamber. Further study could be of interest for performing lysis research using nanospike arrays in a flowing environment. Meriner et al. [17] proposed a rectangular channel (length = 2 cm, width = 2 mm, and height = 100 μm) with arrays of 3D carbon electrodes; each electrode was a cylindrical post with a diameter of 58 μm extending from the bottom to the top wall of the channel, and the minimum spacing between electrodes was also 58 μm. The electrodes were actuated by AC voltages, with a 180° phase shift between neighboring electrodes. Lysis throughput was achieved up to 600 μL/min, at high cell density (10^8^ yeast cells per mL) with 90% efficiency under 130 V_pp_. A voltage of 65 V_pp_ was required for a throughput at 100 μL/min with 98% efficiency.

The sawtooth structure, the MWCNT coating, the nanospike arrays, and the 3D carbon electrode arrays listed above are all designs that aim for producing local high intensity fields for electric lysis. As planar electrodes were employed commonly in microfluidic devices using AC electrokinetics for decades, and it is well-known that the edges of planar electrodes usually generate localized high intensity fields [18,19,20], it is of interest to see if traditional planar electrodes can be employed for effective electrical lysis, which is the primary goal of the present study. 

The theory of electroporation and the mechanisms of electrical lysis were usually attributed to the microscopic interaction between the applied electric field and the lipid bilayers enclosing the cells [21,22,23]. However, the electric field also exerts stress, called the Maxwell stress [24], on the cell from a macroscopic point of view; and a critical stress can always be related to rupturing a material (the cell here). The role of Maxwell stresses on electrical cell lysis was not studied. Thus, the second goal of the present study is to understand the relationship between Maxwell stress and electrical lysis. In particular, the threshold Maxwell stress, which is associated with the irreversible threshold electric field strength of electroporation (*E*_th2_), will be contrasted to the threshold stress of mechanical lysis. 

## 2. Materials and Methods 

Both experiments and theoretical calculations were performed, and their details are as follows.

### 2.1. Experimental

The idea for the design of the proposed electrical lysis device is sketched in Figure 1a. The device was essentially a rectangular microchannel with an electrode array (or two planar interdigitated comb-like electrode structures) built on its bottom wall. The electrodes were actuated by AC voltages with a 180° phase shift between neighboring electrodes. Cells in buffer were pumped through the device continuously using a syringe pump, and it was anticipated that all the cells would be lysed as they passed through the electrode region under proper electrical conditions (applied voltage $V_{0}$, frequency *ω*, electrode width *d*, and spacing *s*), channel geometry (width *a*, height *b*, and length *L*), and volume flow rate *Q*. In order to observe the cell conditions at the inlet (intact cells) and outlet (cells were lysed, or not) simultaneously in a window, the microchannel was curved into a U-shape such that the inlet and outlet were placed side-by-side, as shown in Figure 1b. The bell-shape connectors at both the inlet and outlet were designed for reducing the flow velocity for a clear observation of the flowing cells. The flow velocity could be reduced further using tapered connectors with larger dimensions, as shown in Figure 1c. The widths of the channels were chosen as 50, 100, or 150 μm, with the channel height (also the heights of the inlet/outlet connectors) fixed at 43 μm. Both the electrode spacing and width were chosen to be 50 μm, and there were 52 electrodes on the substrate beneath the U-shape channel. The total length of the channel was, thus, about 1.07 cm, and the medium (cells plus buffer solution) would experience about 52 periods (with “wavelength” *λ* = 2*d* + 2*s*) of electrical excitation as it traveled through the device.

The device was fabricated via standard microelectromechanical system (MEMS) techniques, including photolithography, wet etching, and molding using polydimethylsiloxane (PDMS), similar to those in fabricating a traveling wave dielectrophoretic pump for blood delivery [25]. The electrode array was fabricated with gold (100 nm in thickness) on a glass substrate. A chrome layer (30 nm) was inserted between the gold layer and the glass substrate to improve adhesion. The glass substrate with electrode array served as the bottom wall of the channel and also the device. The other three walls of the channel and inlet/outlet connectors were molded with PDMS and bonded to the glass substrate.

The completed device is shown in Figure 2a. As temperature could crucially affect the present electrical cell lysis due to Joule heating, a temperature sensor was also fabricated on the substrate at the outlet for monitoring the temperature, as shown in Figure 2b, using the same technique as that in fabricating the electrodes for cell lysis. Measurement of the temperature sensor was based on the thermal-resistive effect, and it was calibrated using a commercial infrared-ray temperature sensor. The change of temperature $∆T=T-T_{0}$ was linearly proportional to the change in electrical resistance $R-R_{0}$, as shown in Figure 2c, where $R_{0}$ is the resistance at the reference temperature $T_{0}$. A correlation according to the data in Figure 2c was obtained as:(1)$$∆T=T-T_{0}=\frac{1}{\alpha }\left(\frac{R-R_{0}}{R_{0}}\right),\;\mathrm{with}\;\alpha =0.026\;{}^{\circ}C^{-1}$$

Study of electrical cell lysis was performed here using human red blood cells, mainly because they are important cells for many biomedical applications, and have been studied thoroughly in mechanical lysis. Informed consent procedures for blood donations were approved by the National Taiwan University Institutional Review Board. Blood was collected from adult volunteers and kept in vacutainers, which contained sodium polyanethole sulfonate and acid citrate dextrose additives as anticoagulants. Whole blood diluted 10 times with phosphate-buffered solution (PBS) (containing about 6 × 10^8^ cells per mL, mainly red blood cells) was pumped through the device using a syringe pump at rates of 20–500 μL/hr under an applied AC voltage primarily at 20 V_pp_ and 1 MHz. Such a frequency was chosen for avoiding electrolysis and AC electro-osmosis. Smaller voltages (down to 16 V_pp_) were also applied in the experiment, but the lysis occurred only at sufficiently low flow rates. 

The device was first filled with dilute blood for the experiment. The pump was then switched on for 10 s, which allowed the fluid flow to reach a steady state in the device. The power for the electrodes was turned on next to initiate electric cell lysis. The movements of cells at inlet/outlet (see the recording windows in Figure 1b,c) were observed and recorded simultaneously through a microscope and using a digital camera during the experiment; and the image data (30 pictures per second) were stored in a computer for further analysis. The whole device was mounted on an XYZ Table, such that it could be translated horizontally, say, moving along the channel, if one wished to observe cell lysis inside the channel. 

### 2.2. Theory

Details of the theory and the associated equations are available in the Appendix A, and below is a summary of the results.

#### 2.2.1. Fluid Flow

As the length of the microchannel in the present experiment was of two orders greater than its width and height, and the Reynolds number was on the order of unity, the primary flow in the microchannel was a Poiseuille flow, a fully-developed laminar flow in a rectangular channel, with velocity components [26]:(2)$$u=\frac{4a^{2}}{\mu \pi^{3}}\left(-\frac{dp}{dx}\right)\sum_{i=1,3,5,\ldots }^{\infty }{\left(-1\right)}^{(i-1)/2}\left\{1-\frac{\cosh\left[i\pi \left(z-b/2\right)/a\right]}{\cosh\left[i\pi (b/2)/a\right]}\right\}\frac{\cos\left[i\pi \left(y-a/2\right)/a\right]}{i^{3}},v=0,w=0,$$and volume flow rate: (3)$$Q=\frac{ba^{3}}{12\mu }\left(-\frac{dp}{dx}\right)\left[1-\frac{192a}{\pi^{5}b}\sum_{i=1,3,5,\ldots }^{\infty }\frac{\tanh\left[i\pi (b/2)/a\right]}{i^{5}}\right]$$in terms of the notations in Figure 1a. Here (*u*, *v*, and *w*) are the velocity components in the (*x*, *y*, and *z*) coordinates, *dp/dx* is the pressure gradient along the *x*-direction, and *μ* is the dynamic viscosity of the fluid (which was taken as 1.2 cP here for the mixture of whole blood and PBS solution). The non-zero stress components *τ_zx_* = *τ_xz_* = *μ*∂*u*/∂*z* and *τ_yx_* = *τ_xy_* = *μ*∂*u*/∂*y* can be calculated directly, and the maximum magnitude of the mechanical shear stress, *τ*_max_, occurs at the locations (*x*, 0.5*a*, 0) and (*x*, 0.5*a*, *b*) in Figure 1a. For a given volume flow rate *Q*, *dp*/*dx* can be calculated with Equation (3), and thus *u* through Equation (2), as well as *τ_zx_*, *τ_yx_* and *τ*_max_ through appropriate equations. 

In real situations, the flow field in the channel is not strictly parallel to the channel wall. There are velocity components *v* and *w* in the cross-sectional plane, though they are small in comparison with *u*, due to the entrance effect, the secondary flow generated in the bend [27], the flow generated by Joule heating [19,20], and the local unsteady flow associated with rupture of cells. These transverse flows, especially the last two, were helpful for effective cell lysis in the present continuous flow-through device, as will be discussed later. However, the primary Poiseuille flow in Equations (2) and (3) provided us an adequate estimation, at least on the correct order of magnitude, for the mechanical shear stresses contributed by the fluid flow in the device.

#### 2.2.2. Electric Field and Maxwell Stress Tensor

The electric field, and thus the Maxwell stress tensor, is periodic along the axis of the channel for most of the region in the device except near the inlet and the bend in Figure 1b. The electric potential in the fluid medium (with electrical conductivity ≈ 1 S/m here) is governed by the Laplace equation [28], and it was solved in the region 0≤x≤λ,
0≤y≤a, and 0≤z≤b in Figure 1a, subject to specified AC potentials on the electrodes, insulated boundary conditions at the glass (electrical conductivity ≈ 10^−15^–10^−11^ S/m) and PDMS (electrical conductivity ≈ 10^−15^–10^−11^ S/m) walls, and periodic conditions along the *x*-direction. Such a geometric configuration and boundary conditions suggest that the electric potential, Ψ, is two-dimensional in the *xz*-plane and varies with time, *t*, and it can be written as: (4)Ψ(x,z,t)=Φ(x,z)cos(ωt),with Φ(*x*,*z*) solved numerically using a finite difference approximation and the Gauss–Seidel iterative method. With electric potential known, the electric field, E(x,z,t), can be calculated through:(5)$$\begin{array}{ll} E\left(x,z,t\right) & =E_{x}e_{x}+E_{z}e_{z}=-\nabla \Psi =-\nabla \Phi \cos\left(\omega t\right)=-\frac{\partial \Phi }{\partial x}\cos\left(\omega t\right)e_{x}-\frac{\partial \Phi }{\partial z}\cos\left(\omega t\right)e_{z}\\ & =E_{p}\cos\left(\omega t\right)=E_{\mathrm{px}}\cos\left(\omega t\right)e_{x}+E_{\mathrm{pz}}\cos\left(\omega t\right)e_{z} \end{array}$$with $e_{x}$ and $e_{z}$ representing the unit vectors along the *x* and *z* directions, respectively. The corresponding electric field components *E_x_* and *E_z_* are time-varying functions in the AC field, and their time-averages are zero. We will examine the electric field phasor components, *E*_px_ and *E*_pz_, and the root mean square of the electric field.(6)$$E_{\mathrm{rms}}=\sqrt{\bar{E_{x}^{2}+E_{z}^{2}}}=\sqrt{\frac{\omega }{2\pi }\int_{0}^{2\pi /\omega }(E_{x}^{2}+E_{z}^{2})dt}=\frac{1}{\sqrt{2}}\sqrt{{\left(-\frac{\partial \Phi }{\partial x}\right)}^{2}+{\left(-\frac{\partial \Phi }{\partial z}\right)}^{2}}=\frac{1}{\sqrt{2}}\sqrt{E_{\mathrm{px}}^{2}+E_{\mathrm{pz}}^{2}},$$where the “mean” refers to the average over time for one period (2*π*/*ω*) of electric field variation. 

The Maxwell stress tensor can be calculated according to its definition [24], and the time mean Maxwell stress tensor, expressed in matrix form, is(7)$$T=\left[\begin{array}{ccc} T_{xx} & T_{xy} & T_{xz}\\ T_{yx} & T_{yy} & T_{yz}\\ T_{zx} & T_{zy} & T_{zz} \end{array}\right]=\varepsilon_{0}\varepsilon_{r}\left[\begin{array}{ccc} 0.25\left(E_{\mathrm{px}}^{2}-E_{\mathrm{pz}}^{2}\right) & 0 & 0.5E_{\mathrm{px}}E_{\mathrm{pz}}\\ 0 & -0.25\left(E_{\mathrm{px}}^{2}+E_{\mathrm{pz}}^{2}\right) & 0\\ 0.5E_{\mathrm{pz}}E_{\mathrm{px}} & 0 & 0.25\left(E_{\mathrm{pz}}^{2}-E_{\mathrm{px}}^{2}\right) \end{array}\right],$$where $\varepsilon_{0}=8.854\times 10^{-12}m^{-3}kg^{-1}s^{4}A^{2}$ is the permittivity in vacuum, and $\varepsilon_{r}$ is the relative permittivity of the medium. We had stress components *T_xx_*, *T_zz_*_,_ and *T_xz_* on the *xz*-plane, as well as a compressive component *T_yy_*, for the present two-dimensional electric field in the *xz*-plane. The stresses changed with the orientation of the surface they applied. The time mean principal stresses, *T*_tensile_ (maximum tension) and *T*_compressive_ (maximum compression), and the time mean maximum shear stress, *T*_shear_, following standard textbook of mechanics of materials ([29], for example), can be derived as: (8)$$T_{\mathrm{tensile}}=\frac{1}{2}\varepsilon_{0}\varepsilon_{r}E_{\mathrm{rms}}^{2},\;T_{\mathrm{compressive}}=-\frac{2}{3}\varepsilon_{0}\varepsilon_{r}E_{\mathrm{rms}}^{2}\;,\;\mathrm{and}\;T_{\mathrm{shear}}=\frac{7}{12}\varepsilon_{0}\varepsilon_{r}E_{\mathrm{rms}}^{2}.$$

## 3. Results

### 3.1. Experiments

Figure 3 and Figure 4 show snapshots at different times from Appendix A for the lysis of human red blood cells of cases B1 and T2, respectively, in Table 1. The dilute blood was pumped through the device for more than 10 s before the electric power of the electrodes was turned on. The clock was set to 0 s when the power was on in Figure 3 and Figure 4. The primary difference between these two cases was that they had different inlets/outlets, as shown in Figure 1b,c, respectively, such that the flowing cells could be observed more clearly in case T2 even though its volume flow rate was twice that of case B1. Consider first case B1 in Figure 3. The flow of cells was quite rapid such that we could see only crowded “streaklines” (the black lines) of cells instead of the individual cells clearly. The streakline pattern at the bell-shape outlet region at 0 s appeared essentially the same as that at −10 s, indicating that the mechanical shear and extensive stresses alone from the flow could not induce cell lysis for the present device. After the electric field was applied for 3 s, the streaklines disappeared at the outlet as shown in the figure at 3 s, showing that the cells were lysed. The time interval between the instant when streaklines (cells) first disappeared at the outlet and the instant when the power was switched on was defined here as the time for complete lysis, *t*_CL_, for describing the “efficiency” of cell lysis. A smaller value of *t*_CL_ represented a better efficiency. Note that *t*_CL_ is a parameter defined here for convenience, not a parameter precisely defined technically. A transit time, *t*_tr_, was required for the fluid and cells to flow through the device. Here we chose *t*_tr_ as *L*/*u*_av_, with *u*_av_ the average axial flow speed in the channel, and *L* the channel length. Such a *t*_tr_ was just a characteristic value for the transit time representing an average value. The cells seen at the outlet were roughly *t*_tr_ earlier on average than those in the inlet. *t*_CL_ = 3 s > *t*_tr_ = 1.59 s for case B1, because there were certain amounts of cells moving closer to the wall, and they needed longer times to transit the channel. *t*_tr_ was a parameter averaged over all the cells while *t*_CL_ was a parameter that took into account all the cells through the channel, and *t*_CL_ > *t*_tr_ for all the cases in Table 1. No streaklines (no cells) were observed in the outlet, in comparison with crowded streaklines in the inlet, between 3 s to 10 s when the power of the electrodes was on, showing that cells were lysed continuously as they passed through the device during this time interval. The streaklines (cells) reappeared in the outlet region 1 s (see the figure at 11 s) after the power of electrodes was turned off at 10 s. The streakline pattern at 12 s (2 s after the power was off) recovered and looked similar to that at 0 s. 

Consider next the case T2 in Figure 4 from Appendix A. A tapered inlet/outlet shown in Figure 1c was employed for a stronger flow reduction at the outlet. The result in Figure 4 was qualitatively similar to that in Figure 3, except we needed here a longer time for complete lysis, *t*_CL_ = 10 s in case T2 instead of 3 s in Case B1. The cells were completely lysed continuously between 10–20 s in Figure 4. Note that some cells appeared in the outlet regions shown in the figures at 10 s and 20 s, but they were adhered cells on the wall, as observed clearly in Appendix A. Detailed examination of the cells in the figures at 10 s and 20 s also suggested that most of the cells in these two figures were at the same locations, which implied further that those cells probably adhered to the wall prior to electrical lysis. Also, the number of adhered cells was very few in comparison with the large number of cells passing through the device, as shown in the video. Those adhered cells were not observed when trypsin was added in the dilute blood for the experiment. Similar to Figure 3, the cells reappeared in the outlet shortly (2 s for case T2) after the power had been turned off, as shown in the figure at 22 s in Figure 4.

In summary, effective and continuous electrical cell lysis was demonstrated clearly in the present proposed simple flow-through device using classical planar electrodes, as observed with Figure 3 (Appendix A) and Figure 4 (Appendix A).

Experiments for complete cell lysis subjected to an applied peak-to-peak voltage 20 V_pp_ at 1 MHz are listed in Table 1. Summarized in Table 1 are the channel width *a* (with channel height *b* = 43 μm, and channel length *L* = 1.07 cm holding fixed), the volume flow rates *Q*, the negative pressure gradient driving the flow −*dp/dx* according to Equation (3), the magnitude of the maximum mechanical shear stress *τ*_max_, the average axial speed *u*_av_ = *Q/*(*ab*), the transit time of cells through the channel *t*_tr_ = *L*/*u*_av_, the time for complete lysis *t*_CL__,_ the temperature at the outlet (*T*_out_) when complete lysis was first observed *T*_CL_, and the scale for shear stress *τ*_scale_ = *μ**u*_av_/*b*. The experiments in Table 1 were performed primarily with bell-shape inlets and outlets, except two cases (T1 and T2). Those two cases using a tapered inlet and outlet had larger values of *T*_CL_ in comparison to the corresponding bell-shape cases.

Three channel widths, 50, 100, and 150 μm, were tested with different volume flow rates, as shown in Table 1. Cases with flow rates greater than those listed in Table 1 were also tested, for example, *Q* = 500 μL/hr for the case with *a* = 50 μm, but the cells could only be partially lysed. The magnitude of the maximum mechanical shear stress from the fluid flow, *τ*_max_, in Table 1 was of one to three orders less than the threshold value of mechanical lysis [30,31,32,33]. Also, no cell lysis was observed between −10 s to 0 s in the corresponding videos associated with Figure 3 and Figure 4. Thus, cell lysis due solely to mechanical shear was ruled out in the present device. However, mechanical shear did play certain role when the electric power was on, as will be discussed in Section 4.3.

The time for complete lysis, *t*_CL_, increased as the volume flow rate increased, up to a critical value. Beyond the critical volume flow rate, the cells could only be partially lysed. Further increase of the volume flow rate led to a decrease in the percentage of the cells that could be lysed. Also, *t*_CL_ increased as the channel width increased at a given volume flow rate, primary because of the decrease in cell speed transiting the channel.

All of the above discussions in this section were for cases with applied voltage equaling 20 V_pp_. Experiments at lower applied voltages were also performed, and it was found that the minimum applied peak-to-peak voltage (2*V*_0_) for complete lysis was 16 V_pp_, and the minimum flow rate was about 1 μL/hr in the present device. The map for complete lysis for 16–20 V_pp_ and *u*_av_ = 1.3–51.7 mm/s is shown in Figure 5. Here, the average axial speed was employed, instead of the volume flow rate as a parameter for illustration, to take into account the data from different channel widths. A higher applied voltage was required for a higher flow speed. No cell lysis occurred when the flow speed was zero, as the negative dielectrophoretic force pushed all the cells away from the electrode edges where the local field strength was high; the cells settled on the central part of the electrodes without lysis.

### 3.2. Calculations

In order to understand in better detail electrical cell lysis in the present device, it was required to know some quantitative results of the electric field and Maxwell stress inside the channel. Numerical calculations were performed, and here we considered the case with an applied voltage 2*V*_0_ = 20 V_pp_, frequency = 1 MHz, channel height *b* = 43 μm, and wavelength of electrical excitation *λ* = 2*s* + 2*d* = 100 μm + 100 μm = 200 μm (refer to Figure 1a) as an illustration.

The contours of the electric field phasor components are shown in Figure 6a,b. The field phasor emitted from the electrode between *x* = 75 to 125 μm to the neighboring electrodes between −25 to 25 μm and 175 to 225 μm on the substrate (*z* = 0). The root mean square of the electric field, *E*_rms_, is shown in Figure 6c,d, with the latter showing more clearly the distribution around the value 1.2 × 10^5^ V/m, which is the threshold of the irreversible electroporation, *E*_th2_, according to Ref. [13]. If *E*_th2_ = 1.2 × 10^5^ V/m was a criterion for the threshold of electric cell lysis based on the electric field magnitude, the contour of such a value (the black M-shape curve) in Figure 6d marked the boundary between a region of cell lysis and that without lysis. Cells travelling below the M-shape curve would be lysed provided they were subject to a sufficiently long exposure time. The area below the M-shape curve indicated that about half of the region inside the channel was the lysing region. In order to provide more detailed values of the electric field, variations of *E*_rms_ with *x* at different heights (*z*) were also shown in Figure 6e,f at different scales. The spatial distribution of the electric field is highly non-uniform, with high intensity spike-like regions above the electrode edges located at *x* = 25, 75, 125, and 175 μm. Those spike-like regions, as observed in Figure 6a–e, penetrated several microns into the medium and were two-dimensional in the device. They built up invisible “walls” of high intensity electric field across the channel from *y* = 0 to *a* in the lower part of the device in Figure 1a, tending to intercept the travelling cells and lysing them. 

The two-dimensional “walls” of high intensity electric field generated by the edges of planar electrodes were more effective at intercepting translating cells in comparison with the spike-like high intensity field regions (like a “fence” instead of a “wall”) generated by three-dimensional electrodes in the literature, for example, Ref. [16]. There were four such “walls” in each cycle of electrical excitation; thus, a total of 200 “walls” were present in the channel of the device, as shown in Figure 1b. 

Figure 7 shows the time-average maximum Maxwell shear stress, *T*_shear_, corresponding to the electric field in Figure 6. The results of *T*_shear_ in Figure 7 appeared qualitatively similar to those of *E*_rms_ in Figure 6, as $T_{\mathrm{shear}}=(7/12)\varepsilon_{0}\varepsilon_{r}E_{rms}^{2}$ according to Equation (8), where $\varepsilon_{r}$ was chosen as 77 for the present mixture of whole blood and PBS solution in the calculation. However, the distribution of *T*_shear_ (∝$E_{rms}^{2}$) was even more non-uniform than that of *E*_rms_, and the 200 “walls” of high intensity electric field mentioned above were also “walls” of high Maxwell shear stress. 

It is well-known in the literature that mechanical shear can induce cell lysis [30,31,32,33], and the threshold shear stress for mechanical cell lysis, *T*_th-me-shear_, decreases as the exposure time for shearing increases. For the present device, the exposure time was on the order of 1 s; and *T*_th-me-shear_ for such an exposure time could be estimated to be about 2000 dynes/cm^2^ (or 200 Pa) according to Figure 8 of Ref. [30]. It was expected that the threshold value of cell lysis for shear under an electric field, *T*_th-shear_, should be less than *T*_th-me-shear_, because the cell membrane was weakened by electroporation. By comparing Figure 6d with Figure 7b, the contour of 5.7 Pa in Figure 7b resembled that of 1.2 × 10^5^ V/m in Figure 6d; they marked the same boundary between two regions in the associated figures. If 1.2 × 10^5^ V/m was the threshold electric field magnitude for cell lysis, 5.7 Pa was the threshold Maxwell shear stress, *T*_th-shear_, which separated below the lysed region and above the non-lysed region in the channel. *T*_th-shear_ was just about 3% of *T*_th-me-shear_.

The results of *T*_tensile_ and *T*_compressive_ were also calculated but were not shown here. They were qualitatively the same as those of *T*_shear_ but with different magnitudes. *T*_tensile_ = 6/7 *T*_shear_ and *T*_compressive_ = −8/7 *T*_shear_ according to Equation (8), and, thus, the results in Figure 7 could be employed for *T*_tensile_ or *T*_compressive_ by timing the values with the corresponding factors, 6/7 or −8/7, respectively. In the literature of mechanical cell lysis, the threshold of tensile stress for lysis, *T*_th-me-tensile_, was greater than, but of the same order as, *T*_th-me-shear_, and was reported as 300 Pa [34] and 500 Pa [35], as examples. There was no specific discussion on the threshold of compressive stress for mechanical cell lysis in the literature. However, pressure (compressive stress) on the order of 10 MPa (which is much greater than *T*_th-me-shear_) is required in high-pressure homogenization for large scale cell disruption [3]. Thus, *T*_th-shear_ was considered mainly here, as it was guessed to be less than the associated threshold values of tensile and compressive Maxwell stresses, if there was analogy between electrical (Maxwell) and mechanical stresses.

## 4. Discussion

### 4.1. Thermal or Electrical Lysis

Progressive morphological changes occurred for red blood cells when they were exposed to a temperature of 50 °C [36] or higher, and the red blood cells were completely lysed (ruptured into small pieces) at 55 °C according to microscopic observations of cell changes with gradually increasing temperatures in the present study. The values of *T*_CL_ in Table 1 are far less than 55 °C, indicating that cell lysis of the cases in Table 1 is not due to thermal means, at least for a finite time duration of electrical excitation (until *T*_out_ reaches 55 °C, due to Joule heating). To see this further, the temperatures at the outlet, *T*_out_, were measured continuously to a time sufficiently greater than *t*_CL_ for cases with different channel widths and different volume flow rates, as shown in Figure 8. Also indicated in Figure 8 (the vertical short line segments that intersect the curves) are the times for complete lysis, *t*_CL_, which were shown before in Table 1. 

Consider the case for 300 μL/hr in Figure 8a, the time difference between the time at 55 °C and *t*_CL_ is about 30 s, showing that the cells were electrically lysed at least for a 30 s time duration. Similarly, electrical lysis occurred for more than 60 s for the case of 100 μL/hr in the 50 μm channel in Figure 8a, and it was more than 50 s for the cases with channel widths of 100 μm when the flow rate was below 100 μL/hr, as shown in Figure 8b. The cell membrane could be weakened as the temperature increased [36]. The results in Figure 8 also indicated that the system was still evolving thermally in the experiments, though the flow field and the electric field had already reached their steady states. This is because the mass of the glass substrate and PDMS walls are much larger than the dilute blood mass inside the channel, and a longer time is required for the whole system to reach its thermal steady state. More measurements of temperature along the channel and a detailed thermal analysis of the system are required for a clear understanding of the related problem. In case the medical tests of lysate are sensitive to temperature, attention should be paid for controlling the temperature when using electrical lysis.

### 4.2. Electric Field and Cell Lysis

Whether a cell can be electrically lysed depends on how the electric field (field magnitude and exposure time) it experienced during its journey through the channel. Consider first the field magnitude. The irreversible thresholds of electroporation, *E*_th2_, of human red blood cells studied here can be estimated to be 1.2 × 10^5^ V/m, based on the study of cells moving through an orifice (with 10 μm in height and width and 40 μm in length) under an electro-osmotic flow (with a speed 30 ± 9 μm/s) inside a larger microchannel [13]. As for the exposure time, the transit time for a cell through the orifice is 1.33 s (≈ 40 μm/30 μm/s). However, cells were ruptured into small pieces (lysed) and “disappeared” instantaneously in the photos near the entrance of the orifice (within 1/4 of the orifice length). Thus, the exposure time for electrical lysis could be estimated as 0.33 s (≈ 1.33 s / 4) for the corresponding applied field. Such an exposure time is about the same or less than the values of the transit times, *t*_tr_ (Table 1), of our experiments. Therefore, one needs essentially to examine if the electric field magnitude is less than the threshold value, 1.2 × 10^5^ V/m, inside the present channel.

By examining the numerical values in Figure 6d, the cells were shown to experience a sufficiently high strength field (>*E*_th2_ = 1.2 × 10^5^ V/m) for the region below the M-shape black curve, such that they would be lysed if they traveled in the lower half of the channel. Furthermore, they could be lysed more easily and rapidly if they were closer to the electrode surface, as the field there was much stronger.

For cells initially in the upper half of the channel (the region above the “M-shape black curve” in Figure 6d), it is natural to expect that they will move continuously in the upper half region, as the primary Poiseuille flow is a parallel flow along the *x*-direction. If this occurs, those cells in the upper half will transit the channel and will not be lysed, as the field strength they experience during their journeys is less than *E*_th2_. However, cell lysis did occur in the experiments for cases in Table 1. Thus, there should be a downward cross flow with velocity component, *w*, not equal to zero in the channel in real situations for bringing the cells from the upper half region above the “M-shape black curve” to the lower half region. The cross flows associated with the entrance effect and the secondary flow generated in the bend [27] cannot bring the cells across the mid-plane at *z* = *b*/2.; thus, it is a minor contribution. On the other hand, convection associated with the change of electrical conductivity (*σ*) and permittivity (*ε*) due to localized Joule heating, called the electro-thermally-induced flow [19,20], could make a substantial contribution. The time-average electro-thermal force per unit volume driving the convection in the *xz*-plane can be expressed [20] in terms of the present notations as:(9)$$\bar{f_{E}}=\frac{\varepsilon }{2}\left(\frac{\nabla \varepsilon }{\varepsilon }-\frac{\nabla \sigma }{\sigma }\right)\cdot E_{p}E_{p}-\frac{1}{4}E_{p}\cdot E_{p}\nabla \varepsilon$$under the condition that the applied electric frequency *ω* is much less than *σ**/**ε*, with $\varepsilon =\varepsilon_{0}\varepsilon_{r}$. The change in electric conductivity and permittivity is mainly due to the change in temperature (*T*), and can be approximated as [19,20]: $$\nabla \sigma =\frac{\partial \sigma }{\partial T}\nabla T\;\mathrm{and}\;\nabla \varepsilon =\frac{\partial \varepsilon }{\partial T}\nabla T$$

Equation (9) then becomes:(10)$$\bar{f_{E}}=\frac{\varepsilon }{2}\left(\frac{1}{\varepsilon }\frac{\partial \varepsilon }{\partial T}-\frac{1}{\sigma }\frac{\partial \sigma }{\partial T}\right)\left(\nabla T\cdot E_{p}\right)E_{p}-\frac{\varepsilon }{4}\left(\frac{1}{\varepsilon }\frac{\partial \varepsilon }{\partial T}\right)\left(E_{p}\cdot E_{p}\right)\nabla T$$

For typical applications, $\left(1/\sigma \right)\partial \sigma /\partial T\approx 0.02K^{-1}$ and $\left(1/\varepsilon \right)\partial \varepsilon /\partial T\approx -0.004K^{-1}$ [19,20]. There are two terms on the right-hand side of Equation (10). The first term is in the direction of $E_{p}$ or opposite to it, depending on the whether the sign of $\nabla T\cdot E_{p}=\left(\partial T/\partial x\right)E_{\mathrm{px}}+\left(\partial T/\partial z\right)E_{\mathrm{pz}}$ is negative or positive, respectively. The second term is along the direction of ∇T. The temperature field is set up because of the Joule heating, and it is moderated through thermal diffusion and convection, according to the conservation of energy:(11)$$\rho C_{p}\frac{\partial T}{\partial t}+\rho C_{p}\left(u\frac{\partial T}{\partial x}\right)+\rho C_{p}\left(v\frac{\partial T}{\partial y}\right)+\rho C_{p}\left(w\frac{\partial T}{\partial z}\right)=k\left(\frac{\partial^{2}T}{\partial x^{2}}+\frac{\partial^{2}T}{\partial y^{2}}+\frac{\partial^{2}T}{\partial z^{2}}\right)+\frac{1}{2}\sigma E_{rms}^{2}$$where *t* is the time; *u*, *v*, and *w* are the velocity components along the *x*, *y*, and *z* directions; and *k* is the thermal conductivity of the fluid. Here, Equation (11) is averaged over a cycle of electrical excitation, with the time scale of electrical excitation, 2*π*/*ω*, much less than the thermal time scale of the problem. The two-dimensional temperature field in the *xz*-plane was solved for a model problem, without the transient and convection terms in Equation (11) in Ref. [20], and applied for estimating $\bar{f_{E}}$ using Equation (10); thus, a convective circulation in the *xz*-plane was derived. For the present situation under an applied axial flow and a more complicated electric field, an analytical solution for the temperature field is not available, and a numerical solution is required for further detailed analysis. However, according to the experiment in Figure 8, the outlet temperature, T_out_, is greater than the inlet temperature, which is at room temperature (25 °C) and also the initial temperature in the figure. Thus, a positive temperature gradient, ∂T/∂x>0, was set up globally as a result of the imposed axial flow and Joule heating, although the local temperature gradient could be more complicated. Such a positive temperature gradient incorporated with the electric field shown in Figure 6a,b indicates a downward $\bar{f_{E}}$ in the regions 0 < *x* < 50 μm and 100 < *x* < 150 μm, according to the *z*-component of the first term on the right-hand side of Equation (10). Thus, a cell initially above the “M-shape black curve” in Figure 6d could be brought downward if the fluid drag associated with the downward flow generated by $\bar{f_{E}}$ overcomes the upward negative dielectrophoretic force on the cell. Here, the time-average dielectrophoretic force on a spherical particle is [28]:(12)$$\bar{f_{DEP}}=2\pi \varepsilon R^{3}K_{r}\nabla E_{rms}^{2}$$where *R* is the radius of the cells (assuming spherical here), and *K_r_* is the real part of the Clausius–Mossotti factor. Note that $\bar{f_{E}}$ is the “force per unit volume” on the fluid, and $\bar{f_{DEP}}$ is the “force” on a spherical particle. The velocity scale of the flow generated by $\bar{f_{E}}$ can be estimated from the incompressible Navier–Stokes equation with the electro-thermal force included [20], which is written in terms of the present notations as:(13)$$\rho \left(\frac{\partial u}{\partial t}+u\nabla u\right)=-\nabla p+\mu \nabla^{2}u+\bar{f_{E}}$$where **u** is the fluid velocity, and *p* is the dynamic pressure with the gravitational force absorbed in the pressure term. On writing:(14)$$u=u_{\mathrm{Poiseuille}}+u_{\mathrm{ET}},\;p=p_{\mathrm{Poiseuille}}+p_{\mathrm{ET}},\;\mathrm{with}\;\left|u_{\mathrm{ET}}\right|<<\left|u_{\mathrm{Poiseuille}}\right|$$where $u_{\mathrm{Poiseuille}}$ is the velocity of the imposed Poiseuille flow, with its components shown in Equation (2); $u_{\mathrm{ET}}$ is the velocity of the induced electro-thermal flow; $p_{\mathrm{Poiseuille}}$ is the pressure associated with the Poiseuille flow with $\nabla p_{\mathrm{Poiseuille}}=dp/dx\;e_{x}$ and dp/dx shown in Equation (3); and $p_{\mathrm{ET}}$ is the pressure associated with $u_{\mathrm{ET}}$. By substituting Equation (14) into Equation (13) and subtracting the part of the zeroth order Poiseuille flow, the linearized first order approximation of the electro-thermal flow follows:(15)$$\rho \left(\frac{\partial u_{\mathrm{ET}}}{\partial t}+u\frac{\partial u_{\mathrm{ET}}}{\partial x}\right)=-\nabla p_{\mathrm{ET}}+\mu \nabla^{2}u_{\mathrm{ET}}+\bar{f_{E}}$$

The convection term and the viscous term are of the same order here as the Reynolds number is of order unity using the data in Table 1. Thus, the velocity scale of $u_{\mathrm{ET}}$ (denoted as U_ET_) can be obtained by balancing the viscous term $\mu \nabla^{2}u_{\mathrm{ET}}$ with the electro-thermal force $\bar{f_{E}}$. On the other hand, the velocity scale associated with the dielectrophoretic motion (denoted as U_DEP_) can be obtained by balancing $\bar{f_{DEP}}$ with the Stokes drag on the cell in a stagnant fluid, 6πμRU, where **U** is the cell velocity. Thus,(16)$$\frac{U_{\mathrm{ET}}}{U_{\mathrm{DEP}}}\sim \frac{\frac{s^{2}}{\mu }\left|\bar{f_{E}}\right|}{\frac{1}{6\pi \mu R}\left|\bar{f_{DEP}}\right|}\sim \frac{\frac{s^{2}}{\mu }\frac{\varepsilon }{2}\left(\frac{1}{\varepsilon }\frac{\partial \varepsilon }{\partial T}-\frac{1}{\sigma }\frac{\partial \sigma }{\partial T}\right)\left(\frac{∆T}{L}\right)E_{s}^{2}}{\frac{\varepsilon }{3}\frac{R^{2}}{\mu }K_{r}\frac{E_{s}^{2}}{s}}\approx \frac{3}{2}\left(\frac{1}{\varepsilon }\frac{\partial \varepsilon }{\partial T}-\frac{1}{\sigma }\frac{\partial \sigma }{\partial T}\right)\frac{s^{2}}{R^{2}}\frac{s}{K_{r}}\frac{∆T}{L}\approx 1.68\;$$through a scaling analysis, if $\left|E_{p}\right|\approx E_{rms}$ in Equation (10), $\nabla E_{rms}^{2}\approx E_{rms}^{2}/s$ in Equation (12), $\left(1/\varepsilon \right)\partial \varepsilon /\partial T\approx -0.004\;K^{-1}$, $\left(1/\sigma \right)\partial \sigma /\partial T\approx 0.02\;K^{-1}$, cell radius R≈2.5 μm, electrode spacing s=50 μm, real part of the Clausius–Mossotti factor $K_{r}\approx -0.4$ [25], increase of temperature through the channel ∆T≈10 K according to Figure 8 and Table 1, and channel length L≈1.07 cm. $U_{\mathrm{ET}}$ is of the same order and greater than $U_{\mathrm{DEP}}$, and thus $\bar{f_{E}}$ is indeed a plausible force for bringing the cells downward for lysis. 

In addition to the electro-thermally induced flow, there is also another mechanism that could possibly bring the cells downward from the upper part of the channel, as illustrated below. As the cells are electrically lysed in the lower part of the channel, the unsteady flow associated with rupture of cells is violent, creating local low-pressure regions and inducing suction. Such a phenomenon occurs all the way along the channel, as there are 200 “walls” for intercepting and lysing the cells as discussed before. Thus, it is also suggested that the cross flow for transporting the cells toward the lower half-channel could also be generated by the unsteady flow associated with rupture of cells. A two-phase suspension flow analysis could be helpful, though not easy, for a clearer understanding of such a proposed mechanism.

The downward cross flows for bringing the cells in the upper part to the lower part of the channel for lysis discussed above are inherent in the device. In order to increase the downward cross flow, and thus the efficiency of cell lysis, designed downward flows can also be created using microfluidic means. For example, a helical vertical flow can be generated along the channel axis via fabricating bas-relief structures on the channel wall [37]. Such a design has been employed successfully for enhancing mixing passively in many microfluidic devices. 

### 4.3. Maxwell Shear Stress and Mechanical Shear Stress in Electrical Cell Lysis

According to the discussion in Section 3.2, the threshold Maxwell shear stress (*T*_th-shear_) for electrical lysis of human red blood cell under the present electric field was found to be about 5.7 Pa, which was much less 200 Pa, the threshold of cell lysis for purely mechanical shear stress. Such a large contrast in values, 5.7 versus 200 Pa, is interesting but worth further study. We have, thus, examined the electric field and the corresponding Maxwell stress according to the experimental data on the lysis of human red blood cells in Ref. [13]. The electric field in [13] is a DC field, which is unidirectional along an orifice channel, and can be written as **E** = E_x_**e**_x_, where **e**_x_ is the unit vector along the axis of the orifice. The Maxwell stress components are $T_{xx}=-T_{yy}=-T_{zz}=0.5\varepsilon_{r}\varepsilon_{0}E_{x}^{2}$, with all other off-diagonal components equaling zero. The principal stresses and maximum shear stress (one-half of the difference between two principal stresses) can be found, and(17)$$T_{\mathrm{th}-\mathrm{shear}}=0.5\varepsilon_{0}\varepsilon_{r}E_{\mathrm{th}2}^{2}\approx 4.9\;\mathrm{Pa}\;(\mathrm{for}\;\mathrm{unidirectional}\;\mathrm{electric}\;\mathrm{field}\;\mathrm{in}\;[13])$$where $E_{x}=E_{\mathrm{th}2}=1.2\times 10^{5}\;V/m$ has been applied, and $\varepsilon_{r}=77$. The value 4.9 Pa estimated from the unidirectional electric field is slightly (16%) smaller than, but consistent with, the value 5.7 Pa evaluated for the highly non-uniform two-dimensional field in the present device; this suggests that the threshold value of cell lysis under shear could indeed be reduced dramatically under the action of an electrical field. The difference between 4.9 Pa and 5.7 Pa could be attributed to different applied fields, and unidirectional field in a confined orifice channel in [13] could be more effective for the electroporation of the membrane. 

The time-average maximum Maxwell shear stress *T*_shear_ = 0–5.7 Pa (*T*_th-shear_) in the region above the top M-shape black curve in Figure 7b is on the same order as the maximum mechanical shear stress of the primary Poiseuille flow shown in Table 1 (cases B1–B5). This implies that mechanical shear stress, even of moderate strength, may aid cell lysis under an applied electric field. Thus, the normalized mechanical shear stresses, *τ_zx_* /*τ*_scale_ and *τ_yx_* /*τ*_scale_, were calculated and plotted in Figure 9, with *τ*_scale_ = *μu*_av_/*b* shown in Table 1. The mechanical shear stresses are large near the walls, and are relatively weak in the central part of the channel. On the other hand, the Maxwell shear stress decreases with distance from the electrodes on the bottom wall, and it is weakest at the top wall. Thus, mechanical shear stress could play a more important role near the top wall and the upper parts of the two vertical side walls of the channel in the present device for electrical cell lysis. As the width of the channel increases, a wider region with small *τ_yx_* exists in the central part of the channel, as observed by comparing Figure 9d with Figure 9c; this suggests that the global contribution of mechanical shear to cell lysis decreases as the channel width increases.

The time-average Maxwell stress is a function of *x* and *z,* as shown in Figure 7, but the mechanical shear stress based solely on the Poiseuille flow (assuming that the stresses arising from the cross flows are minor) is steady and two-dimensional in the *yz*-plane, as shown in Figure 9; the combination of them is three-dimensional. If the mechanical stress can be linearly combined with the time-mean Maxwell stress in Equation (7), the time mean total stress is: (18)$$T_{\mathrm{total}}=\left[\begin{array}{ccc} 0.25\varepsilon_{0}\varepsilon_{r}\left(E_{\mathrm{px}}^{2}-E_{\mathrm{pz}}^{2}\right)-p & \tau_{xy} & 0.5\varepsilon_{0}\varepsilon_{r}E_{\mathrm{px}}E_{\mathrm{pz}}+\tau_{xz}\\ \tau_{yx} & -0.25\varepsilon_{0}\varepsilon_{r}\left(E_{\mathrm{px}}^{2}+E_{\mathrm{pz}}^{2}\right)-p & 0\\ 0.5\varepsilon_{0}\varepsilon_{r}E_{\mathrm{pz}}E_{\mathrm{px}}+\tau_{zx} & 0 & 0.25\varepsilon_{0}\varepsilon_{r}\left(E_{\mathrm{pz}}^{2}-E_{\mathrm{px}}^{2}\right)-p \end{array}\right]$$where *p* is the dynamic gage pressure, which can be obtained by integrating –*dp*/*dx* (using the data in Table 1) with *p* = 0 at the outlet (atmospheric pressure there) of the channel, and *τ_yx_* and *τ**_zx_* are the mechanical shear stress components of the Posieuille flow. The principal stresses and the maximum shear stress of **T**_total_ can be obtained using similar procedures as those in the Appendix A, but a concise form as that in Equation (8) for pure Maxwell stresses cannot be derived. A numerical solution is required for the principal stresses for each grid point in the channel. 

However, as an illustration for assessing the relative importance of the mechanical stress to Maxwell stress, the time-mean Maxwell shear stress component *T_zx_* = 0.5*ε*_0_*ε_r_E*_pz_*E*_px_ and the time mean total shear stress component *T_zx_* + *τ_zx_* in Equation (18), at volume flow rates 100 and 400 μL/hr on the vertical mid plane at *y* = 25 μm of the channel, were plotted in Figure 10a–c, and those on the vertical cross sectional planes at *x* = 125 μm were plotted in Figure 10d–f. The total shear stress was modified by the mechanical shear stress at 100 μL/hr and altered substantially at 400 μL/hr, especially in the upper half-channel. The stress near the top wall exceeds the above-mentioned threshold value, 5.7 Pa, in Figure 10c,f, but further study is required as the electric field there is less than *E*_th2_ (see Figure 6d). The threshold shear stress under an electric field should depend on the field strength, as the degree of electroporation is proportional to it. 

### 4.4. Parameters of the Applied Electric Field for Cell Lysis

Numerical results for various parameters of the applied electric field on electrical lysis are shown in Figure 11. Figure 11a is Figure 6d discussed before, and re-plotted here as a reference for comparison. Figure 11b shows that the field strength increased 50% accordingly if the applied voltage increased from 20 to 30 V_pp_. The lysis region can be extended from the bottom to the top wall, and all the cells passing through the channel will experience an electric field exceeding *E*_th2_ for at least half the time during their journeys, even when there is no cross flow in the channel. Lysis performance is surely enhanced. However, as Joule heating equals $0.5\sigma E_{\mathrm{rms}}^{2}$ (the last term of Equation (11)), the temperature increases with the square of the electric field. Most of cells could be lysed thermally instead of electrically in practical applications, and the lysate may not be applicable for medical examination. In such a case, the device should be cooled down by some thermal means, or the PBS buffer (with σ ≈ 1 S/m here) should be replaced by some other buffers with less conductivity, say σ ≈ 0.1 S/m or even lower. Figure 11c shows that the electric field strength could also be increased by about 50% if the electrode spacing is reduced by half while keeping the applied voltage at 20 V_pp_. However, the lysis region is similar to that of Figure 11a and cannot be extended to the top wall; but the effect of Joule heating is increased. Figure 11d shows that we can have similar heating effects as that in Figure 11a, but the lysis region can be extended to the top surface if the channel height is reduced from 43 μm to 30 μm. However, the volume flow rate, and thus the throughput, is reduced when keeping the same average flow speed through the channel. 

### 4.5. Applications

The maximum volume flow rate for complete cell lysis of the present experiments in Table 1 is 400 μL/hr (or 6.67 μL/min), or lysing about 6.7 × 10^4^ cells per second. The throughput in terms of the volume flow rate is just about 1% of that for typical biomedical applications. Comparisons of the throughput of the present device with some typical throughputs in the literature are shown in Table 2. The present results are better than those in [13,14] when the cross-sectional areas (*a* × *b*) of the channels are of the same order. In order to increase the volume flow rate, *Q*, the width of the channel, *a*, was increased from tens of microns to millimeters in [15,17] together with the application of a relatively large applied voltage (40–130 V_pp_). As discussed in Section 4.1, the cells and the buffer could be heated up to 55 °C (the critical temperature for thermal lysis) in one minute (see Figure 8) due to Joule heating at an applied voltage of 20 V_pp_. In contrasting the results of the “first case of [17] with *Q* = 100 μL/min” to those of the “Present work” in Table 2, the ratio of cross sectional area for fluid flow of the former to the latter case is about 46.5, the corresponding ratio for Joule heating is (65/20)^2^ ≈ 10.6, and the corresponding ratio for volume flow rate is 15. If one wishes to increase the volume flow rate of the present device to 100 μL/min, but avoid additional Joule heating (still using 20 V_pp_), one plausible way is to use multiple channels (the “Proposed device” in Table 2) similar to that employed in [14]. For practical applications, the channels need not be curved into a U-shape, as those in Figure 1 and Figure 2, which were designed for demonstration. Instead, parallel straight channels are preferred, and fifteen rectangular channels with 50 μm × 43 μm cross-sections can be easily fabricated on the substrate of Figure 2a with a twice as long electrode array. Such a fifteen-channel device could also lyse a similar number of cells as those in the second case of [17], with *Q* = 600 μL/min in Table 2, as the present device can handle a larger concentration of cells. However, some effort had to be done for arranging similar flow rates through different parallel channels. 

As the irreversible thresholds for electroporation are different for different cells, the present device can also be applied for cell separation, for example, destroying one type of cell (e.g., human red blood cells) while leaving other types of cells (e.g., circulating tumor cells) intact when both cells are forced to flow through the device under a suitably designed electric field. Such a separation procedure could significantly contribute to the detection of circulating tumor cells [38]. A preliminary study was performed [39] by spiking human lung cancer cells, CL1-5, into dilute human blood and letting the mixture flow through a device similar to the present device. The recovered rates of cancer cells were 54%, 90%, and 96% for three tests.

## 5. Conclusions

A simple flow-through device is proposed and demonstrated here successfully for continuous and massive lysis of cells via electricity, using dilute human whole blood in phosphate-buffered saline. The device is essentially a rectangular microchannel with a planar electrode array built on its bottom wall, actuated by AC voltages around 20 V_pp_ with a 180° phase shift between neighboring electrodes, and it can be incorporated easily into other biomedical systems. Electric field and Maxwell stress were calculated for understanding the underlying physics and assessing the performance of electrical lysis in the present device. Besides the background driving flow along the channel, cross flows also exist. They are proposed here primarily as the electro-thermally induced flow and the unsteady flow associated with the rupture of cells, and they are crucial for electrical lysis. The Maxwell shear stress associated with the threshold electric field strength of irreversible electroporation is one order of magnitude less than the threshold mechanical shear stress for lysis, implying that an applied moderate mechanical stress, which occurs in many microfluidic devices, could aid electrical lysis. Some applications of the present method were also discussed.

## Figures and Tables

**Figure 1 micromachines-10-00247-f001:**
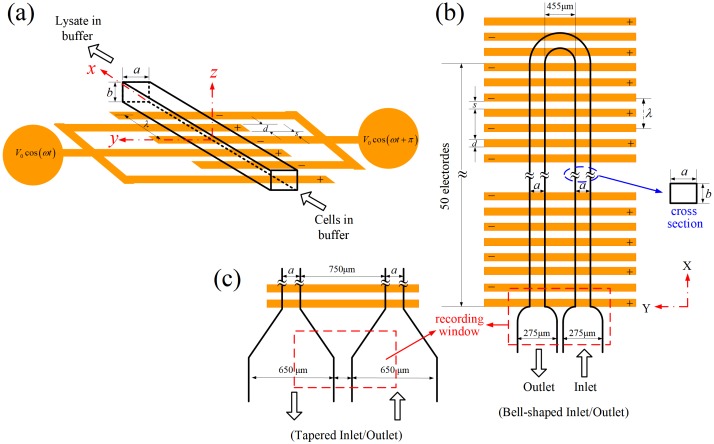
(**a**) Sketch of a simple flow-through device for continuous electrical lysis of cells. (**b**) Top view of the device employed in the present study. The flow phenomenon of cells at the inlet and outlet can be observed and recorded simultaneously in this experimental design. (**c**) An alternative inlet/outlet design for reducing the flow in comparison with that in (**b**), for a clear observation of cell lysis.

**Figure 2 micromachines-10-00247-f002:**
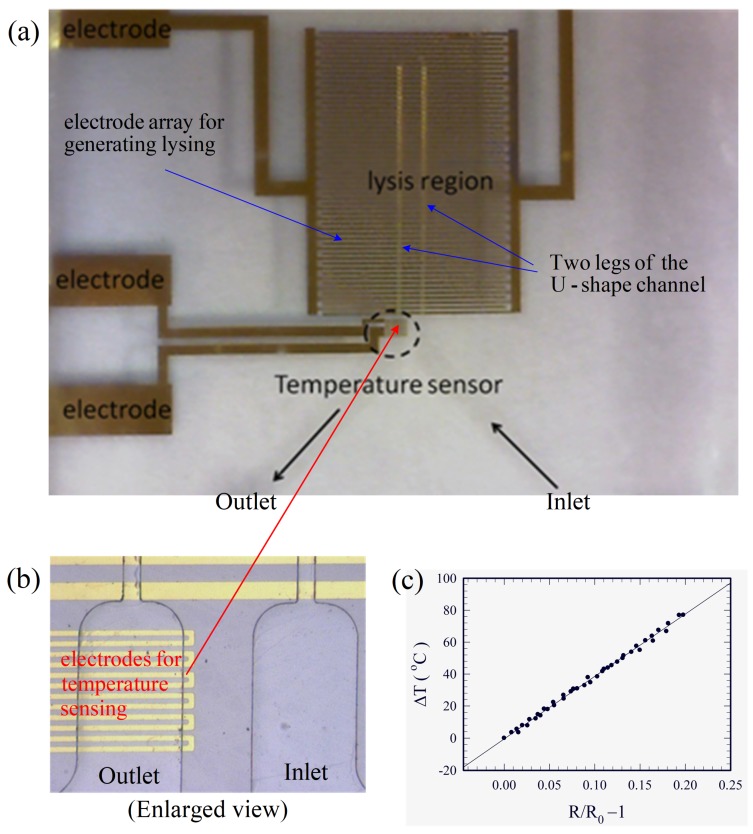
(**a**) The completed device for electrical lysis. (**b**) The temperature sensor built at the exit of the device. (**c**) Variation of the temperature with electric resistance for the temperature sensor.

**Figure 3 micromachines-10-00247-f003:**
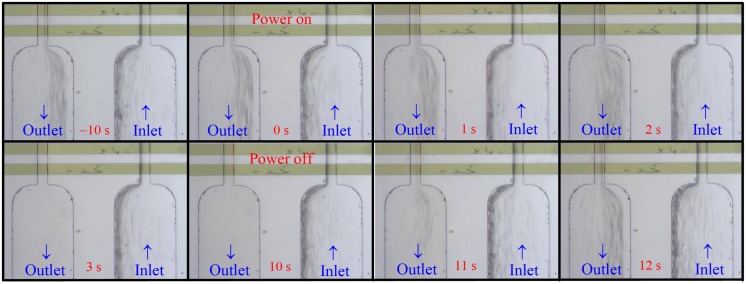
Snapshots at different time instants from Appendix A for case B1 in Table 1 (2*V*_0_ = 20 V_pp_, *a* = 50 μm, *Q* = 50 μL/hr, bell-shape inlet/outlet). No cells were observed (completely lysed) in the outlet region from 3 s to 10 s after the power of the electrodes had been turned on for 3 s, and it clearly shows electrical cell lysis.

**Figure 4 micromachines-10-00247-f004:**
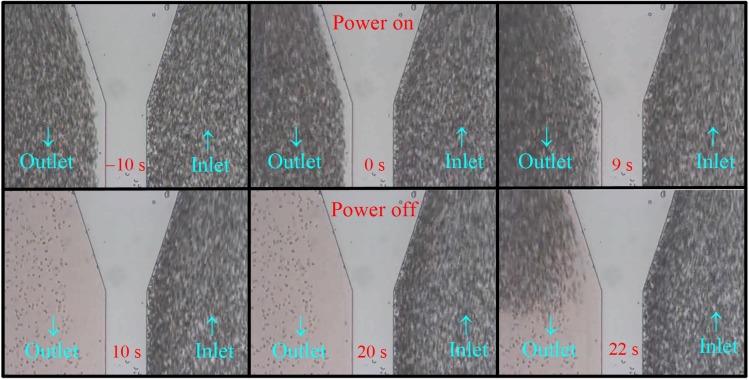
Snapshots at different time instants from Appendix A for case T2 in Table 1 (2*V*_0_ = 20 V_pp_, *a* = 50 μm, *Q* = 100 μL/hr, tapered inlet/outlet). No moving cells were observed (completely lysed) in the outlet region from 10 s to 20 s after the power of the electrodes had been turned on for 10 s.

**Figure 5 micromachines-10-00247-f005:**
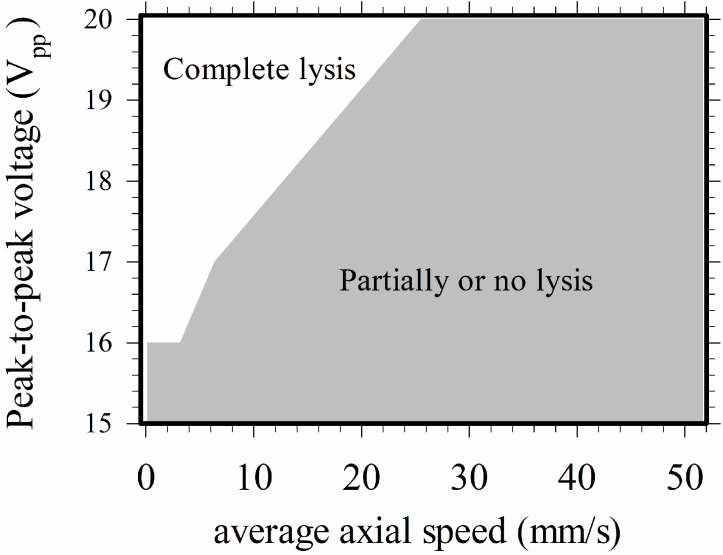
Map for electrical cell lysis in the present device for 16–20 V_pp_ and *u*_av_ = 1.3–51.7 mm/s.

**Figure 6 micromachines-10-00247-f006:**
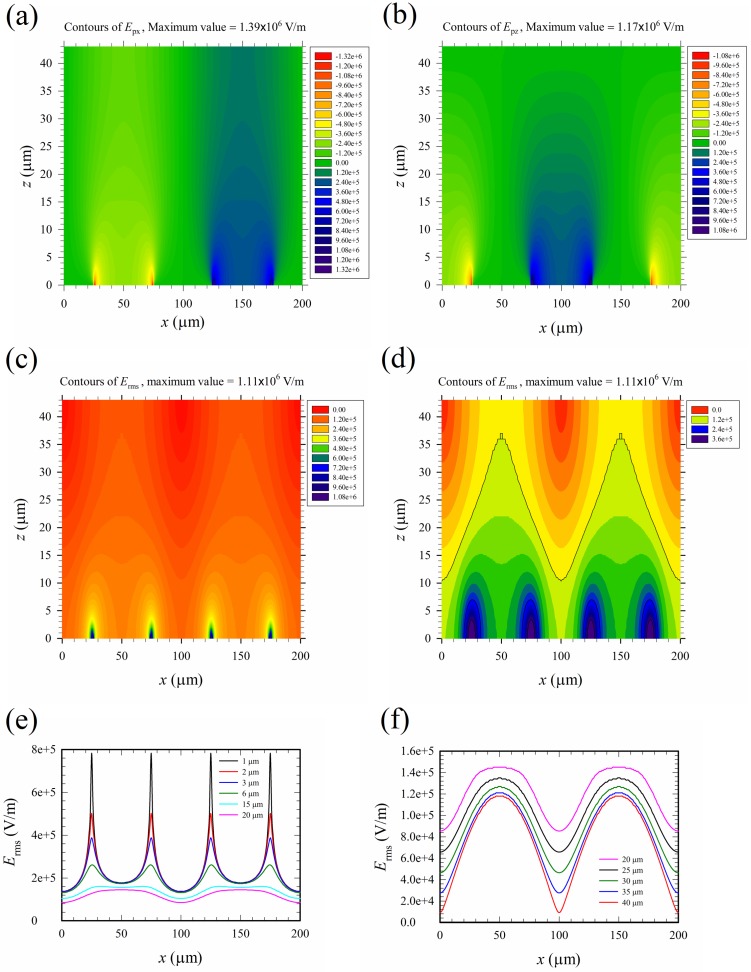
(**a**) Contours of the *x*-component electric field phasor, *E*_px_. (**b**) Contours of the *z*-component electric field phasor, *E*_pz_. (**c**) Contours of the root mean square of the electric field, *E*_rms_ (full range). (**d**) Contours of *E*_rms_ (partial range) for a clear picture of values around 1.2 × 10^5^ V/m, the irreversible threshold value for electroporation, *E*_th2_. The black curves starting from above in (**d**) are the major contours of values 1.2 × 10^5^ (the black M-shape curve), 2.4 × 10^5^, and 3.6 × 10^5^ V/m (four half ellipses enclosing the electrode edges), respectively. There are five minor contours between any two major contours, with a 2 × 10^4^ V/m difference in value between any two minor contours, for example, the values of the minor contours above the M-shape curve are 1.0 × 10^5^, 0.8 × 10^5^, 0.6 × 10^5^, 0.4 × 10^5^ and 0.2 × 10^5^, respectively, starting from below (from yellow to red colors). (**e**) and (**f**) Variations of *E*_rms_ with *x* at different heights (*z*). The applied voltage is 20 V_pp_.

**Figure 7 micromachines-10-00247-f007:**
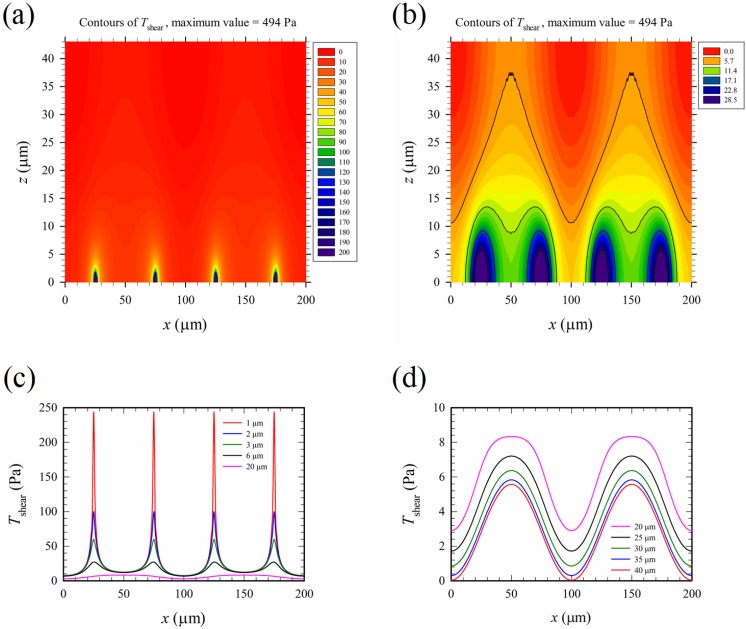
(**a**) Contours of the maximum Maxwell shear stress, *T*_shear_ (full range). (**b**) Contours of *T*_shear_ (partial range) for a clear picture of values around 5.7 Pa, which corresponds to *E*_rms_ = *E*_th2_ = 1.2 × 10^5^ in Figure 6d. The black curves starting from above in (**b**) are the major contours of values 5.7, 11.4, 17.1, 22.8, and 28.5 Pa, respectively (similar convention as that in Figure 6). There are five minor contours between any two major contours, with a 0.95 Pa difference in values between any two minor contours. (**c**) and (**d**) Variations of *T*_shear_ with *x* at different heights (*z*). The applied voltage is 20 V_pp_.

**Figure 8 micromachines-10-00247-f008:**
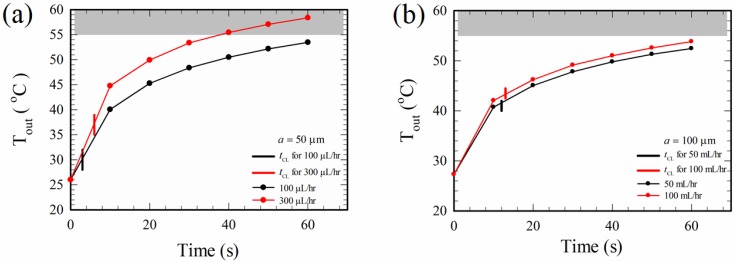
Variations of outlet temperatures, *T*_out_, with time for different flow rates and channel widths: (**a**) *a* = 50 μm, and (**b**) *a* = 100 μm. The vertical short line intersecting each curve marks the time for complete cell lysis of the corresponding case. The grey region for temperature greater than 55 °C is the region for thermal lysis of red blood cells.

**Figure 9 micromachines-10-00247-f009:**
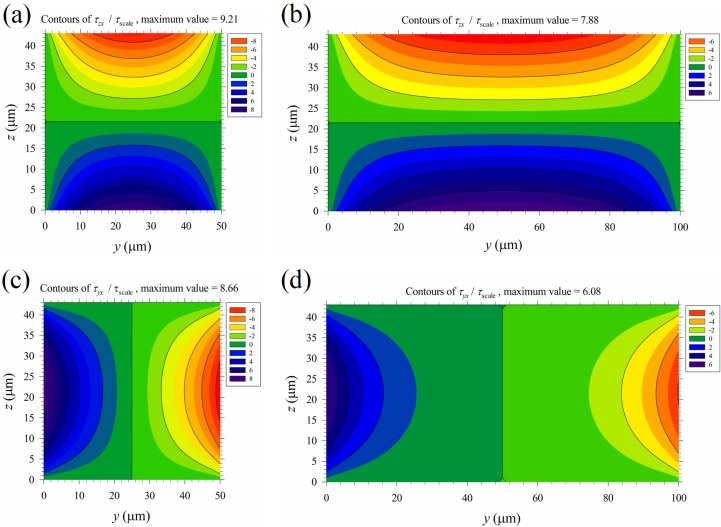
The distributions of the normalized mechanical shear stresses of the Posieuille flow. (**a**) *τ_zx_* /*τ*_scale_, *a* = 50 μm. (**b**) *τ_zx_* /*τ*_scale_, *a* = 100 μm. (**c**) *τ_yx_* /*τ*_scale_, *a* = 50 μm. (**d**) *τ_yx_* /*τ*_scale_, *a* = 100 μm. The black curves show the major contours (starting from the region near the walls), with values the same as those in the color bars. There is one minor contour between any two major contours. Here *τ*_scale_ = *μu*_av_/*b*, with values for different cases shown in Table 1.

**Figure 10 micromachines-10-00247-f010:**
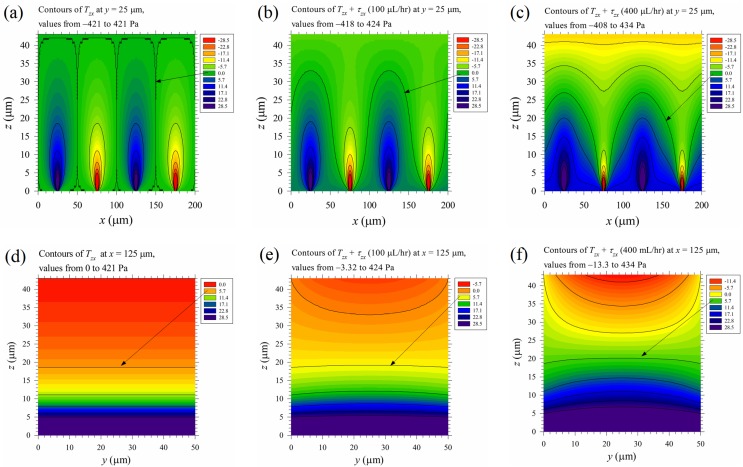
The relative importance between the time-mean Maxwell shear stress component *T_zx_* and the mechanical shear stress component, *τ**_zx_*. (**a**–**c**) are results on the vertical mid plane at *y* = 25 μm, and (**d**–**f**) are results on the vertical cross sectional plane at *x* = 125 μm. (**a**) and (**d**) are the pure Maxwell stresses; (**b**) and (**e**) are the total stresses with volume flow rates of 100 μL/hr, while (**c**) and (**f**) are those at 400 μL/hr. The applied voltage is 20 V_pp_. The sign conventions of the contours are similar to those in Figure 7b.

**Figure 11 micromachines-10-00247-f011:**
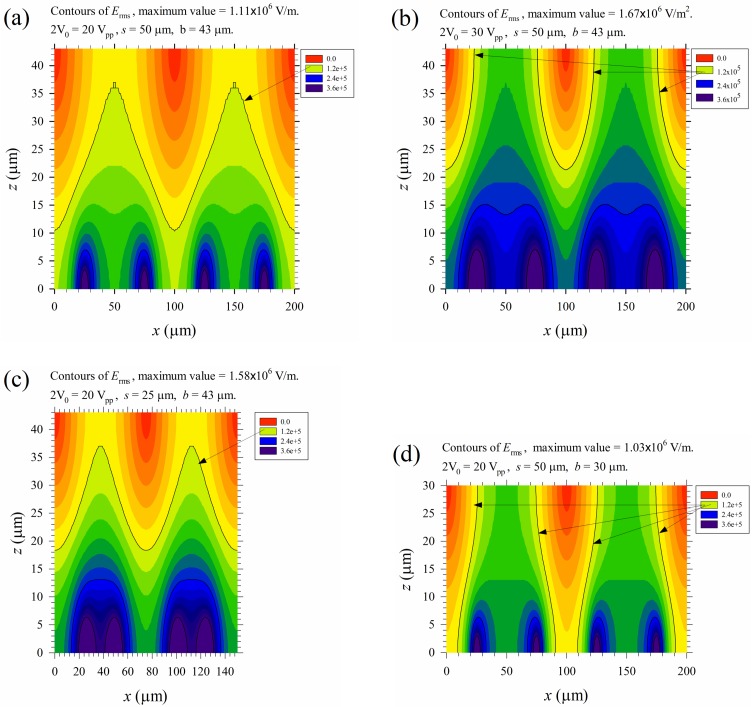
Distributions of *E*_rms_ for different cases. (**a**) 2*V*_0_ = 20 V_pp_, *s* = 50 μm, and *b* = 43 μm; (**b**) 2*V*_0_ = 30 V_pp_, *s* = 50 μm, and *b* = 43 μm; (**c**) 2*V*_0_ = 20 V_pp_, *s* = 25 μm, and *b* = 43 μm; and (**d**) 2*V*_0_ = 20 V_pp_, *s* = 50 μm, and *b* = 30 μm. Here, 2*V*_0_ is the applied peak-to-peak voltage, *s* is the electrode spacing, and *b* is the channel height. The contours for *E*_rms_ = *E*_th2_ = 1.2 × 10^5^ V/m are indicated by arrows.

**Table 1 micromachines-10-00247-t001:** Summary of the experiments for complete lysis of human blood cells under 20 V_pp_ and 1 MHz.

Case^1^	*a* (μm)	*Q* (μL/hr)	−*dp*/*dx* (Pa/m)	*τ*_max_ (Pa)	*u*_av_ (mm/s)	*t*_tr_ (s)	*t*_CL_ (s)	*T*_CL_ (^o^C)	*τ*_scale_ (Pa)	Inlet/Outlet
B1	50	50	1.04 × 10^5^	1.66	6.46	1.59	3	28	0.18	Bell-shape
B2	50	100	2.08 × 10^5^	3.32	12.9	0.8	3	30	0.36	Bell-shape
B3	50	200	4.15 × 10^5^	6.64	25.8	0.4	4	32	0.72	Bell-shape
B4	50	300	6.22 × 10^5^	9.97	38.8	0.26	6	37	1.08	Bell-shape
B5	50	400	8.29 × 10^5^	13.3	51.7	0.2	7	40	1.44	Bell-shape
T1	50	50	1.04 × 10^5^	1.66	6.46	1.59	10	38	0.18	Tapered
T2	50	100	2.08 × 10^5^	3.32	12.9	0.8	10	40	0.38	Tapered
B11	100	25	1.73 × 10^4^	0.36	1.62	6.36	10	39	0.023	Bell-shape
B12	100	50	3.44 × 10^4^	0.71	3.23	3.18	12	42	0.045	Bell-shape
B13	100	100	6.94 × 10	1.42	6.46	1.59	13	43	0.09	Bell-shape
B21	150	20	8.18 × 10^3^	0.18	0.86	11.9	18	42	0.008	Bell-shape
B22	150	40	1.63 × 10^4^	0.35	1.72	5.97	18	42	0.016	Bell-shape
B23	150	60	2.46 × 10^4^	0.53	2.58	3.98	18	42	0.024	Bell-shape

^1^ Case names with “B” and “T” refer to cases with bell-shape and tapered inlets/outlets, respectively.

**Table 2 micromachines-10-00247-t002:** Comparisons of throughputs among several devices.

References	*a* × *b* × *L*	*Q*	Voltage	*N* _cell_	Cells	Lysis rate	*N*
Lee & Cho [13]	10 × 10 × 40	0.18	50 V (DC)	NA	RBC	100%	1
Lu et al. [14]	30 × 50 × 16200	1	8.5 V_pp_	10^5^–10^6^	HT-29	74%	4
Shahini & Yeow [15]	1000 × 75 × 10000	167	40 V_pp_	NA	*E. Coli*	100%	1
Shahini & Yeow [15]	1000 × 75 × 10000	333	75 V_pp_	NA	*E. Coli*	100%	1
Mernier et al. [17]	2000 × 100 × 20000*	100	65 V_pp_	10^8^	Yeast	98%	1
Mernier et al. [17]	2000 × 100 × 20000*	600	130 V_pp_	10^8^	Yeast	90%	1
Present work	50 × 43 × 10700	6.67	20 V_pp_	6 × 10^8^	RBC	100%	1
Proposed device	50 × 43 × 10700	100	20 V_pp_	6 × 10^8^	RBC	100%	15

*a* = channel width (μm), *b* = channel height (μm), *L* = channel length (μm), *Q* = Volume flow rate (μL/min), *N*_cell_ = number concentration of cells (#/mL), *N* = Numbers of channels, RBC = Human red blood cells, and HT-29 = Human colorectal cancer cells. * Half of the cross-sectional area, *a* × *b*, was blocked by electrode posts.

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
