# Peer review of "A Continuous Flow-through Microfluidic Device for Electrical Lysis of Cells"

_micromachines, 2019, doi:10.3390/mi10040247_

Round 1

Reviewer 1 Report

The paper presents a very interesting and highly sought-after practical application for dielectrophoresis, which I believe should be published forthwith, subject to some minor corrections.

My principal concern is that at 23 pages, the paper is far too long (it has a feel of a PhD thesis chapter, where the student is keen to get recognition for all their work).  The results are interesting and the treatise on the origin of the effect is neat, but there are too many things and the message gets lost as a consequence.  I would give serious consideration to downsizing the theory and simulation sections; they (and the Appendix) could easily be moved to a Supplementary page if needs be.  This allows the main paper to be tightened up and focused more on the results, which are ultimately more important.  Note that this doesn’t mean removing all the theory or simulations, but just keeping the key results. 

I think one aspect of the design which ought to be addressed is that of relatively low flow rate.  200ul/hr is not much when biologists - who are the ultimate users of this technology, rather than "lab on a chip people" - work in the ml/min regime.  The cell throughput (ca. 6e8/ml) is excellent, or ca. 80,000 cells/second at 500ul/hr (some x4 faster than high-throughput FACS) but biologists tend to work in higher volumes.  This isn't a criticism, but it would be good to see the authors discuss how the technology might be used in a wider context (e.g. a standalone device) rather than just as part of a wider microsystem.

It would also be good to know if this DEP system could also perform simultaneous electroportation and separation - i.e. just destroying one population.  

I would suggest the description of the device, currently on page 3, be moved to a separate section.  This could be completely separate or form part of the Materials and Methods, but shouldn’t be in the Introduction as this is principally intended to provide background, not new work.  This is particularly the case for Figure 1, which currently reads as if it’s part of pre-existing literature rather than the design used here.

There are a few typos, e.g. line 21 hyphen missing (should be “half-channel”), line 29 application should be plural, and so on.  Reference 11 also seems to have merged into the end of reference 10.  A thorough proofread would be good.

However, despite these I believe this represents an important application with widespread advantages, for which the authors should be congratulated.

Reviewer 2 Report

The manuscript has been written very well and it is suitable for publication in Micromachines.

Author Response

Thank you for the reviewer's comment. We have checked the typos and spells carefully in the revised manuscript, and made the corrections.